# STATISTICAL INFERENCE FOR FISHER MARKET EQUILIBRIUM

**Luofeng Liao, Yuan Gao, Christian Kroer**
Department of Industrial Engineering and Operations Research
Columbia University
{ll3530,yg2541,christian.kroer}@columbia

## ABSTRACT

Statistical inference under market equilibrium effects has attracted increasing attention recently. In this paper we focus on the specific case of linear Fisher markets. They have been widely use in fair resource allocation of food/blood donations and budget management in large-scale Internet ad auctions. In resource allocation, it is crucial to quantify the variability of the resource received by the agents (such as blood banks and food banks) in addition to fairness and efficiency properties of the systems. For ad auction markets, it is important to establish statistical properties of the platform's revenues in addition to their expected values. To this end, we propose a statistical framework based on the concept of infinite-dimensional Fisher markets. In our framework, we observe a market formed by a finite number of items sampled from an underlying distribution (the "observed market") and aim to infer several important equilibrium quantities of the underlying long-run market. These equilibrium quantities include individual utilities, social welfare, and pacing multipliers. Through the lens of sample average approximation (SAA), we derive a collection of statistical results and show that the observed market provides useful statistical information of the long-run market. In other words, the equilibrium quantities of the observed market converge to the true ones of the long-run market with strong statistical guarantees. These include consistency, finite sample bounds, asymptotics, and confidence. As an extension we discuss revenue inference in quasilinear Fisher markets.

## 1 INTRODUCTION

In a Fisher market there is a set of $n$ buyers that are interested in buying goods from a distinct seller. A market equilibrium (ME) is then a set of prices for the goods, along with a corresponding allocation, such that demand equals supply.

One important application of market equilibrium (ME) is fair allocation using the competitive equilibrium from equal incomes (CEEI) mechanism (Varian, 1974). In CEEI, each individual is given an endowment of faux currency and reports her valuations for items; then, a market equilibrium is computed, and the items are allocated accordingly. The resulting allocation has many desirable properties such as Pareto optimality, envy-freeness and proportionality. For example, Fisher market equilibrium has been used for fair work allocation, impressions allocation in certain recommender systems, course seat allocation and scarce computing resources allocation; see Appendix A for an extensive overview.

Despite numerous algorithmic results available for computing Fisher market equilibria, to the best of our knowledge, no statistical results were available for quantifying the randomness of market equilibrium. Given that CEEI is a fair and efficient mechanism, such statistical results are useful for quantifying variability in CEEI-based resource allocation. For example, for systems that assign blood donation to hospitals and blood banks (McElfresh et al., 2020), or donated food to charities in different neighborhoods (Aleksandrov et al., 2015; Sinclair et al., 2022), it is crucial to quantify the variability of the amount of resources (blood or food donation) received by the participants (hospitals or charities) of these systems as well as the variability of fairness and efficiency metrics of interest

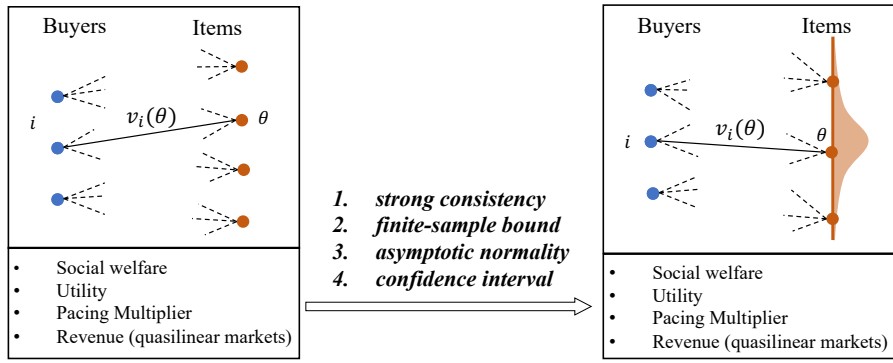

**Figure 1:** Our contributions. Left panel: a Fisher market with a finite number of items. Right panel: a Fisher market with a continuum of items. Middle arrow: this paper provides various forms of statistical guarantees to characterize the convergence of observed finite Fisher market (left) to the long-run market (right) when the items are drawn from a distribution corresponding to the supply function in the long-run market.

in the long run. Making statistical statements about these metrics is crucial for both evaluating and improving these systems.

In addition to fair resource allocation, statistical results for Fisher markets can also be used in revenue inference in Internet ad auction markets. While much of the existing literature uses expected revenue as performance metrics, statistical inference on revenue is challenging due to the complex interaction among bidders under coupled supply constraints and common price signals. As shown by Conitzer et al. (2022a), in budget management through repeated first-price auctions with pacing, the optimal pacing multipliers correspond to the "prices-per-utility" of buyers in a quasilinear Fisher market at equilibrium. Given the close connection between various solution concepts in Fisher market models and first-price auctions, a statistical framework enables us to quantify the variability in long-run revenue of an advertising platform. Furthermore, a statistical framework would also help answer other statistical questions such as the study of counterfactuals and theoretical guarantees for A/B testing in Internet ad auction markets.

For a detailed survey on related work in the areas of statistical inference, applications of Fisher market models, and equilibrium computation algorithms, see Appendix A.

Our contributions are as follows.

**A statistical Fisher market model**. We formulate a statistical estimation problem for Fisher markets based on the continuous-item model of Gao and Kroer (2022). We show that when a finite set of goods are sampled from the continuous model, the observed ME is a good approximation of the long-run market. In particular, we develop consistency results, finite-sample bounds, central limit theorems, and asymptotically valid confidence interval for various quantities of interests, such as individual utility, Nash social welfare, pacing multipliers, and revenue (for quasilinear Fisher markets).

**Technical challenges**. In developing central limit theorems for pacing multipliers and utilities in Fisher markets (Theorem 5), we note that the dual objective is potentially not twice differentiable. This is a required condition, which is common in the sample average approximation or M-estimation literature. We discover three types of market where such differentiability is guaranteed. Moreover, the sample function is not differentiable, which requires us to verify a set of stochastic differentiability conditions in the proofs for central limit theorems. Finally, we achieve a fast statistical rate of the empirical pacing multiplier to the population pacing multiplier measured in the dual objective by exploiting the local strong convexity of the sample function.

**Notation.** For a sequence of events $A_n$ we define the set limit by $\liminf_{n\to\infty} A_n = \bigcup_{n\geq 1}\bigcap_{j\geq n} A_j = \{A_t \text{ eventually}\}$ and $\limsup_{n\to\infty} A_n = \bigcap_{n\geq 1}\bigcup_{j\geq n} A_j = \{A_t \text{ i.o.}\}$. Let $[n] = \{1,\ldots,n\}$. We use $1_t$ to denote the vector of ones of length $t$ and $e_j$ to denote the vector with one in the $j$-th entry and zeros in the others. For a sequence of random variables $\{X_n\}$, we say $X_n = O_p(1)$ if for any $\epsilon > $ there exists a finite $M_\epsilon$ and a finite $N_\epsilon$ such that $\mathbb{P}(|X_n| > M_\epsilon) < \epsilon$ for all $n \geq N_\epsilon$. We say $X_n = O_p(a_n)$ if $X_n/a_n = O_p(1)$. We use subscript for indexing buyers

and superscript for items. If a function $f$ is twice continuously differentiable at a point $x$, we say $f$ is $C^2$ at $x$.

## 2 PROBLEM SETUP

### 2.1 THE ESTIMANDS

Following Gao and Kroer (2022), we consider a Fisher market with $n$ buyers (individuals), each having a budget $b_i > 0$ and a (possibly continuous) set of items $\Theta$. We let $L^p$ (and $L^p_+$, resp.) denote the set of (nonnegative, resp.) $L^p$ functions on $\Theta$ w.r.t the integrating measure $\mu$ for any $p \in [1, \infty]$ (including $p = \infty$). For example, one could take $\Theta = [0, 1]$ and $\mu =$ the Lebesgue measure on $[0, 1]$. The item *supplies* are given by a function $s \in L^\infty_+$, i.e., item $\theta \in \Theta$ has supply $s(\theta)$. The *valuation* for buyer $i$ is a function $v_i \in L^1_+$, i.e., buyer $i$ has valuation $v_i(\theta)$ for item $\theta \in \Theta$. For buyer $i$, an *allocation* of items $x_i \in L^\infty_+$ gives a utility of

$$u_i(x_i) := \langle v_i, x_i \rangle := \int_\Theta v_i(\theta) x_i(\theta) \, \mathrm{d}\mu(\theta),$$

where the angle brackets are based on the notation of applying a bounded linear functional $x_i$ to a vector $v_i$ in the Banach space $L^1$ and the integral is the usual Lebesgue integral. We will use $x \in (L^\infty_+)^n$ to denote the aggregate allocation of items to all buyers, i.e., the concatenation of all buyers' allocations. The *prices* of items are modeled as $p \in L^1_+$. The price of item $\theta \in \Theta$ is $p(\theta)$. Without loss of generality, we assume a unit total supply $\int_\Theta s \, \mathrm{d}\mu = 1$. We let $S(A) := \int_A s(\theta) \, \mathrm{d}\mu(\theta)$ be the probability measure induced by the supply $s$.

Imagine there is a central policymaker that sets the prices $p(\cdot)$ of items $\Theta$. Upon receiving the price signal, the buyer $i$ maximizes his utility $\langle v_i, x_i \rangle$ subject to the budget constraint $\langle p, x_i \rangle \leq b_i$. He would demand a bundle of items coming from his demand set

$$D_i(p) := \arg\max_{x_i} \{\langle v_i, x_i \rangle : x_i \in L^\infty_+, \langle p, x_i \rangle \leq b_i\}.$$

Of course, due to the supply constraint, if prices are too low, there will be a shortage in supply. On the other hand, if prices are too high, a surplus occurs. Market equilibrium is the case when items are sold out exactly.

**Definition 1** (The long-run market equilibrium). *The market equilibrium (ME) $\mathscr{ME}(b, v, s)$ is an allocation-utility-price tuple $(x^*, u^*, p^*) \in (L^\infty_+)^n \times \mathbb{R}^n_+ \times L^1_+$ such that the following holds. (i) Supply feasibility and market clearance: $\sum_i x_i^* \leq s$ and $\langle p^*, s - \sum_i x_i^* \rangle = 0$. (ii) Buyer optimality: $x_i^* \in D_i(p^*)$ and $u^* = \langle v_i, x_i \rangle$ for all $i$.*

Linear Fisher market equilibrium can be characterized by convex programs. We state the following result from Gao and Kroer (2022) which establishes existence and uniqueness of market equilibrium, and more importantly the convex program formulation of the equilibrium. We define the Eisenberg-Gale (EG) convex programs which as we will see are dual to each other.

$$\max_{x \in L^\infty_+(\Theta), u \geq 0} \left\{ \mathrm{NSW}(u) := \sum_{i=1}^n b_i \log(u_i) \,\Big|\, u_i \leq \langle v_i, x_i \rangle \; \forall i \in [n], \; \sum_{i=1}^n x_i \leq s \right\}, \quad \text{(P-EG)}$$

$$\min_{\beta > 0} \left\{ H(\beta) := \int_\Theta \Big( \max_{i \in [n]} \beta_i v_i(\theta) \Big) S(\mathrm{d}\theta) - \sum_{i=1}^n b_i \log \beta_i \right\}. \quad \text{(P-DEG)}$$

Concretely, the optimal primal variables in Eq. (P-EG) corresponds to the set of equilibrium allocations $x^*$ and the unique equilibrium utilities $u^*$, and the unique optimal dual variable $\beta^*$ of Eq. (P-DEG) relates to the equilibrium through

$$u_i^* = b_i / \beta_i^*, \quad p^*(\theta) = \max_i \beta_i^* v_i(\theta), \quad x_i^*(\theta) > 0 \text{ only if } i \in \arg\max_i \beta_i^* v_i(\theta).$$

We call $\beta^*$ the *pacing multiplier*. Note equilibrium allocations might not be unique but equilibrium utilities and prices are unique. Given the above equivalence result, we use $(x^*, u^*)$ to denote both the equilibrium and the optimal variables. Another feature of linear Fisher market is full budget extraction: $\int p^* \, \mathrm{d}S = \sum_{i=1}^n b_i$; we discuss quasilinear model in Appendix F.

We formally state the first-order conditions of infinite-dimensional EG programs and its relation to first-price auctions in Fact 1 in appendix. Also, we remark that there are two ways to specify the valuation component in this model: the functional form of $v_i(\cdot)$, or the distribution of values $v : \Theta \to \mathbb{R}^n_+$ when view as a random vector. More on this in Appendix E.

We are interested in estimating the following quantities of the long-run market equilibrium. (1) **Individual utilities** at equilibrium, $u_i^*$. (2) **Pacing multipliers** $\beta_i^* = b_i/u_i^*$. Pacing multiplier has a two-fold interpretation. Second, through the equation $p^*(\theta) = \max_i \beta_i^* v_i(\theta)$, $\beta$ can also be interpreted as the *pacing policy*[1] employed by the buyers in first-price auctions. In our context, buyer $i$ produces a bid for item $\theta$ by multiplying the value by $\beta_i$, then the item price is determined via a first-price auction. This connection is made precise in Conitzer et al. (2022a) from a game-theoretic point of view. The pacing multiplier $\beta$ serves as the bridge between Fisher market equilibrium and first price pacing equilibria in auction games (Conitzer et al., 2022a) and has important usage in budget management in online auctions. (3) The (logarithm of) **Nash social welfare** (NSW) at equilibrium $\text{NSW}^* := \text{NSW}(u^*)$. NSW measures total utility of the buyers and, when used as an optimization objective, is able to promote fairness better than the social welfare, that is, the sum of buyer utilities (Caragiannis et al., 2019). Intuitively, NSW incentivizes more balancing of buyer utilities. Revenue inference for quasilinear model is discussed in Appendix F[2].

**Mapping model to concrete applications.** It is well-known that Fisher market is a useful mechanism for fair and efficient resource allocation. More recently, it is also shown to be intimately related to first-price auctions (Conitzer et al., 2022a). When modeling ad auction platforms, buyers' individual utilities reflect, for example, the values generated to advertisers, measured in terms of click rates, conversion rates, or other revenue metrics. A confidence interval on this quantity can be provided by the ad platform to the advertisers to better inform advertisers' decision-making. Pacing multiplier, as the ratio between the bid and the value of items, reflects the advertisers' bidding strategy. A confidence interval on this quantity could help the ad platform predict its clients bidding behavior and suggest budget management strategy to clients. Finally, Nash social welfare measures the efficiency and fairness of the whole ad platform. We can also use Fisher market to model resource allocation systems in internet companies. For example, in a job recommendation platform, we can model it as a market where we distribute viewer's attention to job post creators. In this context, social welfare measures generally the efficiency of the job recommendation system. Individual utilities track how satisfied the job post creators are with the extent to which their job posts are being recommended.

## 2.2 THE DATA

Assume we are able to observe a market formed by a finite number of items. We let $\gamma = \{\theta_1, \ldots, \theta_t\} \subset \Theta^t$ be a set of items sampled i.i.d. from the supply distribution $S$. We let $v_i(\gamma) = \left(v_i(\theta^1), \ldots, v_i(\theta^t)\right)$ denote the valuation for agent $i$ of items in the set $\gamma$. For agent $i$, let $x_i = (x_i^1, \ldots, x_i^t) \in \mathbb{R}^t$ denote the fraction of items given to agent $i$. With this notation, the total utility of agent $i$ is $\langle x_i, v_i(\gamma) \rangle$. Similar to the long-run market, we assume the observed market is at equilibrium, which we now define.

**Definition 2** (Observed Market Equilibrium). *The market equilibrium $\mathcal{ME}^\gamma(b, v, \mathsf{s})$ given the item set $\gamma$ and the supply vector $\mathsf{s} \in \mathbb{R}^t_+$ is an allocation-utitlity-price tuple $(x^\gamma, u^\gamma, p^\gamma) \in (\mathbb{R}^n_+)^n \times \mathbb{R}^n_+ \times \mathbb{R}^t_+$ such that the following holds. (i) Supply feasibility and market clearance: $\sum_{i=1}^n x_i^\gamma \leq \mathsf{s}$ and $\langle p^\gamma, 1_t - \sum_{i=1}^n x_i^\gamma \rangle = 0$. (ii) Buyer optimality: $x_i^\gamma \in D_i(p^\gamma)$ and $u_i^\gamma = \langle v_i(\gamma), x_i \rangle$ for all $i$, where (overloading notations)*

$$D_i(p) := \arg\max_{x_i}\{\langle v_i(\gamma), x_i \rangle : x_i \geq 0, \langle p, x_i \rangle \leq b_i\}$$

*is the demand set given the prices and the buyer's budget.*

---

[1]In the online budget management literature, pacing means buyers produce bids for items via multiplying his value by a constant.

[2] In a linear Fisher market the budgets are extracted fully, i.e., $\int p^* \, dS = \sum_i b_i$ in the long-run market and $\sum_{\tau=1}^t p^{\gamma,\tau} = \sum_i b_i$ in the observed market (see Appendix E), and therefore there is nothing to infer about revenue in this case. However, in the quasilinear utility model where buyer's utility function is $u_i(x) = \langle x - p, v_i \rangle$, buyers have the incentive to retain money and therefore one needs to study the statistical properties of revenues.

Assume we have access to $(x^\gamma, u^\gamma, p^\gamma)$ along with the budget vector $b$, where $(x^\gamma, u^\gamma, p^\gamma) \in \mathscr{ME}^\gamma(b, v, \frac{1}{t}1_t)$ is the market equilibrium (we explain the scaling of $1/t$ in Appendix E). Note the budget vector $b$ and value functions $v = \{v_i(\cdot)\}_i$ are the same as those in the long-run ME. We emphasize two high-lights in this model of observation.

**No convex program solving.** The quantities observed are natural estimators of their counterparts in the long-run market, and so we do not need to perform iterative updates or solve optimization problems. One interpretation of this is that the actual computation is done when equilibrium is reached via the utility maximizing property of buyers; the work of computation has thus implicitly been delegated to the buyers.

For finite-dimensional Fisher market, it is well-known that the observed market equilibrium $\mathscr{ME}^\gamma(b, v, \frac{1}{t}1_t)$ can be captured by the following sample EG programs.

$$\max_{x \geq 0, u \geq 0} \left\{ \mathrm{NSW}(u) \;\middle|\; u_i \leq \langle v_i(\gamma), x_i \rangle \;\forall i, \sum_{i=1}^n x_i^\tau \leq \tfrac{1}{t}1_t \;\forall \tau \right\}, \tag{S-EG}$$

$$\min_{\beta > 0} \left\{ H_t(\beta) := \frac{1}{t}\sum_{\tau=1}^t \max_{i \in [n]} \beta_i v_i(\theta^\tau) - \sum_{i=1}^n b_i \log \beta_i \right\}. \tag{S-DEG}$$

We list the KKT conditions in Appendix E. Completely parallel to the long-run market, optimal solutions to Eq. (S-EG) correspond to the equilibrium allocations and utilities, and the optimal variable $\beta^\gamma$ to Eq. (S-DEG) relates to equilibrium prices and utilities through $u_i^\gamma = b_i/\beta_i^\gamma$, $p^{\gamma,\tau} = \max_i \beta_i^\gamma v_i(\theta^\tau)$ and $x_i^{\gamma,\tau} > 0$ only if $i \in \arg\max_i \beta_i^\gamma v_i(\theta^\tau)$. By the equivalence between market equilibrium and EG programs, we use $u^\gamma$ and $x^\gamma$ to denote the equilibrium and the optimal variables. Let $\mathrm{NSW}^\gamma := \mathrm{NSW}(u^\gamma) = \sum_{i=1}^n b_i \log u_i^\gamma$. All budgets in the observed market is extracted, i.e., $\sum_{\tau=1}^t p^{\gamma,\tau} = \sum_{i=1}^n b_i$.

## 2.3 Dual Programs: Bridging Data and the Estimands

Given the convex program characterization, a natural idea is to study the concentration behavior of observed market equilibria through these convex programs. We focus on the dual programs Eqs. (S-DEG) and (P-DEG) because they are defined in a fixed dimension, and that the constraint set is also fixed.

Define the sample function $F = f + \Psi$, where $f(\beta, \theta) = \max_i \{v_i(\theta)\beta_i\}$, and $\Psi(\beta) = -\sum_{i=1}^n b_i \log \beta_i$; the function $f$ is the source of non-smoothness, while $\Psi$ provides local strong convexity. Our approach is studying concentration of the convex programs in the sense that as $t$ grows

$$\min_{\beta > 0} H_t(\beta) = \frac{1}{t}\sum_{\tau=1}^t F(\beta, \theta^\tau) \quad \text{``} \Longrightarrow \text{''} \quad \min_{\beta > 0} H(\beta) = \mathbb{E}[F(\beta, \theta)].$$

The local strong convexity of the dual objective motivates us to do the analysis work in the neighborhood of the optimal solution $\beta^*$. In particular, the function $x \mapsto -\log x$ is not strongly convex on the positive reals, but it is on any compact subset. By working on a compact subset, we can exploit strong convexity of the dual objective and obtain better theoretical results. It is known (Fact 1) that $\underline{\beta}_i \leq \beta_i^* \leq \bar{\beta}$ where $\underline{\beta}_i = b_i/\int v_i \, \mathrm{d}S$ and $\bar{\beta} = \sum_{i=1}^n b_i/\min_i \int v_i \, \mathrm{d}S$. Define the compact set $C := \prod_{i=1}^n \left[\underline{\beta}_i/2, 2\bar{\beta}\right] \subset \mathbb{R}^n$, which must be a neighborhood of $\beta^*$. Moreover, for large-enough $t$ we further have $\beta^\gamma \in C$ with high probability (c.f. Lemma 1).

**Blanket assumptions.** Recall the total supply in the long-run market is one: $\int s \, \mathrm{d}\mu = 1$. Assume the total item set produce one unit of utility in total, i.e., $\int v_i s \, \mathrm{d}\mu = 1$. Suppose budges of all buyers sum to one, i.e., $\sum_{i=1}^n b_i = 1$. Let $\underline{b} := \min_i b_i$. Note the previous budget normalization implies $\underline{b} \leq 1/n$. Finally, for easy of exposition, we assume the values are bounded $\sup_\Theta v_i(\theta) < \bar{v}$, for all $i$. By the normalization of values and budgets, we know $\underline{\beta}_i = b_i$ and $\bar{\beta} = 1$.

## 3 CONSISTENCY AND FINITE-SAMPLE BOUNDS

In this section we introduce several natural empirical estimators based on the observed market equilibrium, and show that they satisfy both consistency and high-probability bounds. Below we state the consistency results; the formal version can be found in Appendix B.

**Theorem 1** (Consistency, informal). *The Nash social welfare, pacing multiplier and approximate equilibria in the observed market are strongly consistent estimators of their counterparts in the long-run market.*

**High Probability Bounds**  Next, we refine the consistency results and provide finite sample guarantees. We start by focusing on Nash social welfare and the set of approximate market equilibria. The convergence of utilities and pacing multiplier will then be derived from the latter result.

**Theorem 2.** *For any failure probability $0 < \eta < 1$, let $t \geq 2\bar{v}^2 \log(4n/\eta)$. Then with probablity greater than $1 - \eta$, it holds $|\mathrm{NSW}^\gamma - \mathrm{NSW}^*| \leq O(1)\bar{v}\big(\sqrt{n \log((n+\bar{v})t)} + \sqrt{\log(1/\eta)}\big)t^{-1/2}$ where $O(1)$ hides only constants. Proof in Appendix I.*

Theorem 2 establishes a convergence rate $|\mathrm{NSW}^\gamma - \mathrm{NSW}^*| = \tilde{O}_p(\bar{v}\sqrt{n}t^{-1/2})$. The proof proceeds by first establishing a pointwise concentration inequality and then applies a discretization argument.

**Theorem 3** (Concentration of Approximate Market Equilibrium). *Let $\epsilon > 0$ be a tolerance parameter and $\alpha \in (0,1)$ be a failure probability. Then for any $0 \leq \delta \leq \epsilon/2$, to ensure $\mathbb{P}\big(C \cap \mathscr{B}^\gamma(\delta) \subset C \cap \mathscr{B}^*(\epsilon)\big) \geq 1 - 2\alpha$ it suffices to set*

$$t \geq O(1)\bar{v}^2 \min\left\{\frac{1}{\underline{b}\epsilon}, \frac{1}{\epsilon^2}\right\}\left(n \log\left(\frac{16(2n+\bar{v})}{\epsilon-\delta}\right) + \log\frac{1}{\alpha}\right), \tag{1}$$

*where the set $C = \prod_{i=1}^n [\beta_i/2, 2\bar{\beta}]$, and $O(1)$ hides only absolute constants. Proof in Appendix J.*

By construction of $C$ we know $\beta^* \in C$ holds, and so $C \cap \mathscr{B}^*(\epsilon)$ is not empty. By Lemma 1 we know that for $t$ sufficiently large, $\beta^\gamma \in C$ with high probability, in which case the set $C \cap \mathscr{B}^\gamma(\delta)$ is not empty.

**Corollary 1.** *Let $t$ satisfy Eq. (1). Then with probability $\geq 1 - 2\alpha$ it holds $H(\beta^\gamma) \leq H(\beta^*) + \epsilon$.*

By simply taking $\delta = 0$ in Theorem 3 we obtain the above corollary. More importantly, it establishes the fast statistical rate $H(\beta^\gamma) - H(\beta^*) = \tilde{O}_p(t^{-1})$ for $t$ sufficiently large, where we use $\tilde{O}_p$ to ignore logarithmic factors. In words, when measured in the population dual objective where we take expectation w.r.t. the item supply, $\beta^\gamma$ converges to $\beta^*$ with the fast rate $1/t$. This is in contrast to the usual $1/\sqrt{t}$ rate obtained in Theorem 2, where $\beta^\gamma$ is measured in the sample dual objective. There the $1/\sqrt{t}$ rate is the best obtainable.

By the strong-convexity of dual objective, the containment result can be translated to high-probability convergence of the pacing multipliers and the utility vector.

**Corollary 2.** *Let $t$ satisfy Eq. (1). Then with probability $\geq 1 - 2\alpha$ it holds $\|\beta^\gamma - \beta^*\|_2 \leq \sqrt{\frac{8\epsilon}{\underline{b}}}$ and $\|u^\gamma - u^*\|_2 \leq \frac{4}{\underline{b}}\sqrt{8\epsilon/\underline{b}}$.*

We compare the above corollary with Theorem 9 from Gao and Kroer (2022) which establishes the convergence rate of the stochastic approximation estimator based on dual averaging algorithm (Xiao, 2010). In particular, they show that the average of the iterates, denoted $\beta_{\mathrm{DA}}$, enjoys a convergence rate of $\|\beta_{\mathrm{DA}} - \beta^*\|_2^2 = \tilde{O}_p\big(\frac{\bar{v}^2}{\underline{b}^2}\frac{1}{t}\big)$, where $t$ is the number of sampled items. The rate achieved in Corollary 2 is $\|\beta^\gamma - \beta^*\|_2^2 = \tilde{O}_p\big(\frac{n\bar{v}^2}{\underline{b}^2}\frac{1}{t}\big)$ for $t$ sufficiently large. We see that our rate is worse off by a factor of $n$. And yet our estimates are produced by the strategic behavior of the agents without any extra computation at all. Moreover, in the computation of the dual averaging estimator the knowledge of values $v_i(\theta)$ is required, while again $\beta^\gamma$ can be just observed naturally.

# 4 ASYMPTOTICS AND INFERENCE

## 4.1 ASYMPTOTICS AND ANALYTICAL PROPERTIES OF THE DUAL OBJECTIVE

In this section we derive asymptotic normality results for Nash social welfare, utilities and pacing multipliers. As we will see, a central limit theorem (CLT) for Nash social welfare holds under basically no additional assumptions. However, the CLTs of pacing multipliers and utilities will require twice continuous differentiability of the population dual objective $H$, with a nonsingular Hessian matrix. We present CLT results under such a premise, and then provide three sufficient conditions under which $H$ is $C^2$ at the optimum.

**Theorem 4** (Asymptotic Normality of Nash Social Welfare). *It holds that*

$$\sqrt{t}(\text{NSW}^\gamma - \text{NSW}^*) \xrightarrow{\text{d}} \mathcal{N}(0, \sigma^2_{\text{NSW}}) , \qquad (2)$$

*where* $\sigma^2_{\text{NSW}} = \int_\Theta (p^*)^2 \, dS(\theta) - \left( \int_\Theta p^* \, dS(\theta) \right)^2 = \int_\Theta (p^*)^2 \, dS(\theta) - 1$. *Proof in Appendix K.*

To present asymptotics for $\beta$ and $u$ we need a bit more notation. Let $\Theta_i(\beta) := \{\theta \in \Theta : v_i(\theta)\beta_i \geq v_k(\theta)\beta_k, \forall k \neq i\}$, i.e., the *potential* winning set of buyer $i$ when the pacing multiplier are $\beta$. Let $\Theta_i^* := \Theta_i(\beta^*)$. We will see later that if the dual objective is sufficiently smooth at $\beta^*$, then the winning sets, $\Theta_i^*, i \in [n]$, will be disjoint (up to a measure-zero set). Define the *variance of winning values* for buyer $i$ as follows

$$\Omega_i^2 = \int_{\Theta_i^*} v_i^2(\theta) \, dS(\theta) - \left( \int_{\Theta_i^*} v_i(\theta) \, dS(\theta) \right)^2 .$$

**Theorem 5** (Asymptotic Normality of Individual Behavior). *Assume $H$ is $C^2$ at $\beta^*$ with non-singular Hessian matrix $\mathscr{H} = \nabla^2 H(\beta^*)$. Then $\sqrt{t}(\beta^\gamma - \beta^*) \xrightarrow{\text{d}} \mathcal{N}(0, \Sigma_\beta)$ and $\sqrt{t}(u^\gamma - u^*) \xrightarrow{\text{d}} \mathcal{N}(0, \Sigma_u)$, where $\Sigma_\beta = \mathscr{H}^{-1}\text{Diag}(\Omega_i^2)\mathscr{H}^{-1}$ and $\Sigma_u = \text{Diag}(-b_i/(\beta_i^*)^2)\mathscr{H}^{-1}\text{Diag}(\Omega_i^2)\mathscr{H}^{-1}\text{Diag}(-b_i/(\beta_i^*)^2)$. Proof in Appendix K.*

In Theorem 5 we require a strong regularity condition: twice differentiability of $H$, which seems hard to interpret at first sight. Next we derive a set of simpler sufficient conditions for the twice differentiability of the dual objective. Intuitively, the expectation operator will smooth out the kinks in the piecewise linear function $f(\cdot, \theta)$; even if $f$ is non-smooth, it is reasonable to hope the expectation counterpart $\bar{f}$ is smooth, facilitating statistical analysis.

First we introduce notation for characterizing smoothness of $\bar{f}$. Define the gap between the highest and the second-highest bid under pacing multiplier $\beta$ by

$$\epsilon(\beta, \theta) := \max\{\beta_i v_i(\theta)\} - \text{secondmax}\{\beta_i v_i(\theta)\} , \qquad (3)$$

here secondmax is the second-highest entry; e.g., $\text{secondmax}([1, 1, 2]) = 1$. When there is a tie for an item $\theta$, we have $\epsilon(\beta, \theta) = 0$. When there is no tie for an item $\theta$, the gap $\epsilon(\beta, \theta)$ is strictly positive. Let $G(\beta, \theta) \in \partial f(\beta, \theta)$ be an element in the subgradient set. The gap function characterizes smoothness of $f$: $f(\cdot, \theta)$ is differentiable at $\beta \Leftrightarrow \epsilon(\beta, \theta)$ is strictly positive, in which case $G(\beta, \theta) = \nabla_\beta f(\beta, \theta) = e_{i(\beta,\theta)}v_{i(\beta,\theta)}$ with $e_i$ being the $i$-th unit vector and $i(\beta, \theta) = \arg\max_i \beta_i v_i(\theta)$. When $f(\cdot, \theta)$ is differentiable at $\beta$ a.s., the potential winning sets $\{\Theta_i(\beta)\}_i$ are disjoint (up to a measure-zero set).

**Theorem 6** (First-order differentiability). *The dual objective $H$ is differentiable at a point $\beta$ if and only if*

$$\epsilon(\beta, \theta)^{-1} < \infty, \quad \text{for $S$-almost every $\theta$} . \qquad \text{(NO-TIE)}$$

*When Eq. (NO-TIE) holds, $\nabla \bar{f}(\beta) = \mathbb{E}[G(\beta, \theta)]$. Proof and further technical remarks in Appendix L.*

Given the neat characterization of differentiability of dual objective via the gap function $\epsilon(\beta, \theta)$, it is then natural to explore higher-order smoothness, which was needed for some asymptotic normality results. We provide three classes of markets whose dual objective $H$ enjoys twice differentiability.

**Theorem 7** (Second-order differentiability, Informal). *If any one of the following holds, then $H$ is $C^2$ at $\beta^*$. (i) A stronger form of Eq. (NO-TIE) holds, e.g., $\mathbb{E}[\epsilon(\beta, \epsilon)^{-1}]$ or $\text{ess sup}_\theta\{\epsilon(\beta, \theta)^{-1}\}$ is finite in a neighborhood of $\beta^*$. (ii) The distribution of $v = (v_1, \ldots, v_n) : \Theta \to \mathbb{R}_+^n$ is smooth enough. (iii) $\Theta = [0, 1]$ and the valuations $v_i(\cdot)$'s are linear functions.*

## 4.2 INFERENCE

In this section we discuss constructing confidence intervals for Nash social welfare, the pacing multipliers, and the utilities. We remark that the observed NSW, $\text{NSW}^\gamma$, is a negatively-biased estimate of the NSW, $\text{NSW}^*$, of the long-run ME, i.e., $\mathbb{E}[\text{NSW}^\gamma] - \text{NSW}^* \leq 0$.[3] Moreover, it can be shown that, when the items are i.i.d. $\mathbb{E}[\min H_t] \leq \mathbb{E}[\min H_{t+1}]$ using Proposition 16 from Shapiro (2003). Monotonicity tells us that increasing the size of market produces on average less biased estimates of the long-run NSW.

To construct a confidence interval for Nash social welfare one needs to estimate the asymptotic variance. We let $\hat{\sigma}^2_{\text{NSW}} := \frac{1}{t}\sum_{\tau=1}^{t}\big(F(\beta^\gamma, \theta^\tau) - H_t(\beta^\gamma)\big)^2 = \big(\frac{1}{t}\sum_{\tau=1}^{t}(p^{\gamma,\tau})^2\big) - 1$. where $p^{\gamma,\tau}$ is the price of item $\theta^\tau$ in the observed market. We emphasize that in the computation of the variance estimator $\hat{\sigma}^2_{\text{NSW}}$ one does not need knowledge of values $\{v_i(\theta^\tau)\}_{i,\tau}$. All that is needed is the equilibrium prices $p^\gamma = (p^{\gamma,1}, \ldots, p^{\gamma,t})$ of the items. Given the variance estimator, we construct the confidence interval $[\text{NSW}^\gamma \pm z_{\alpha/2}\frac{\hat{\sigma}_{\text{NSW}}}{\sqrt{t}}]$, where $z_\alpha$ is the $\alpha$-th quantile of a standard normal. The next theorem establishes validity of the variance estimator.

**Theorem 8.** *It holds that* $\hat{\sigma}_{\text{NSW}} \xrightarrow{\text{P}} \sigma^2_{\text{NSW}}$. *Given* $0 < \alpha < 1$, *it holds that* $\lim_{t\to\infty}\mathbb{P}\big(\text{NSW}^* \in [\text{NSW}^\gamma \pm z_{\alpha/2}\hat{\sigma}_{\text{NSW}}/\sqrt{t}]\big) = 1 - \alpha$. *Proof in Appendix M.*

Estimation of the variance matrices for $\beta$ and $u$ is more complicated. The main difficulty lies in estimating the inverse Hessian matrix. Due to the non-smoothness of the sample function, we cannot exchange the twice differential operator and expectation, and thus the plug-in estimator, i.e., the sample average Hessian, is a biased estimator for the Hessian of the population function in general. In Appendix D we provide a brief discussion of variance estimation under the following two simplified scenarios. First, in the case where $\mathbb{E}[\epsilon(\beta, \theta)^{-1}] < \infty$ holds in a neighborhood of $\beta^*$, which we recall is a stronger form Eq. (NO-TIE), we prove that a plug-in type variance estimator is valid. Second, if we have knowledge of $\{v_i(\theta^\tau)\}_{i,\tau}$, then we give a numerical difference estimator for the Hessian which is consistent.

## 5 GUIDE TO PRACTITIONERS

We close our theoretical development with practical instructions on how to construct confidence interval with results presented so far. Revenue inference is only well-defined in the context of quasilinear market, which we explore in Appendix F.

**Inference on NSW.** Recall $b_i$ is the budget of agent $i$ and let $u_i^\gamma$ be the observed utility. Let $\text{NSW}^\gamma = \sum_{i=1}^{n} b_i \log u_i^\gamma$ be the NSW in the observed market. This will be a good estiamte of the limit NSW and as the number of items grows, it converges quickly at a rate of $1/\sqrt{t}$. To construct a CI we need to estimate the asymptotic variance. Let $\hat{\sigma}^2_{\text{NSW}} = \frac{1}{t}\sum_{\tau=1}^{t}(p^{\gamma,\tau} - \bar{p}^\gamma)^2$ where $p^{\gamma,\tau}$ is the price of item $\theta^\tau$ and $\bar{p}^\gamma$ is the average price. In words, $\hat{\sigma}^2_{\text{NSW}}$ is the variance of prices. Then a $(1-\alpha)$ CI for the NSW in the long-run market will be $[\text{NSW}^\gamma \pm z_{\alpha/2}\hat{\sigma}_{\text{NSW}}/\sqrt{t}]$, where $z_\alpha$ is the $(1-\alpha)$-th quantile of a standard normal.

**Inference on $\beta^*$ and $u^*$: simplified inference with a bid-gap condition.** Suppose in the ad auction platform one observes that for each item the winning bid is always higher than the swecond bid by some amount, then we could use the following inference procedure. For a precise statement of the condition see Theorem 10. When such a condition holds, the variance expression in Theorem 5 simplifies. Let $\hat{\Omega}_i$ be the variance of item utility that are allocated to buyer $i$, i.e., $\hat{\Omega}_i^2 := \frac{1}{t}\sum_{\tau=1}^{t}(tu_i^{\gamma,\tau} - u_i^\gamma)^2$, where $u_i^{\gamma,\tau} = x_i^{\gamma,\tau}v_i(\theta^\tau)$ is the utility buyer $i$ obtains from item $\theta^\tau$. Define the asymptotic variance estimators $\hat{\Sigma}_\beta = \text{Diag}(\{\hat{\Omega}_i^2(\beta_i^\gamma)^4/b_i^2\})$ and $\hat{\Sigma}_u = \text{Diag}(\{\hat{\Omega}_i^2\})$. The confidence regions for $\beta^*$ and $u^*$ are ellipsoids that center around $\beta^\gamma$ and $u^\gamma$. Concretely, $\text{CR}_\beta = \beta^\gamma + (\chi_{n,\alpha}/\sqrt{t})\hat{\Sigma}_\beta^{1/2}\mathbb{B}$ and the confidence region for $u^*$ is $\text{CR}_u = u^\gamma + (\chi_{n,\alpha}/\sqrt{t})\hat{\Sigma}_u^{1/2}\mathbb{B}$, where $\chi_{k,\alpha}$ is the $(1-\alpha)$-th quantile of a chi-square with degree $k$ and $\mathbb{B}$ is the unit ball in $\mathbb{R}^n$.

**Inference on $\beta^*$ and $u^*$: a robust approach.** Under fairly general setting, which is justified in Section 4.1, the following inference procedure is valid. Choose a smoothing level $\eta_t$, to be

---

[3]Note $\mathbb{E}[\text{NSW}^\gamma] - \text{NSW}^* = \mathbb{E}[\min_\beta H_t(\beta)] - H(\beta^*) \leq \min_\beta \mathbb{E}[H_t(\beta)] - H(\beta^*) = 0$.

chosen roughly of order $\eta_t = \omega(1/\sqrt{t})$, we construct the following matrix $\hat{\mathscr{H}}$ whose $(i,j)$-th entry is $(\hat{\mathscr{H}})_{ij} := \frac{1}{4\eta_t^2}(H_t(\beta^\gamma + \eta_t(e_i + e_j)) - H_t(\beta^\gamma + \eta_t(-e_i + e_j)) - H_t(\beta^\gamma + \eta_t(e_i - e_j)) + H_t(\beta^\gamma + \eta_t(-e_i - e_j)))$. This will be an estimate of the Hessian matrix of the dual objective. Here $H_t$ defined in Eq. (S-EG) needs only values of the realized items but not the value functions $v_i(\cdot)$'s. Let $\hat{\Sigma}_\beta = \hat{\mathscr{H}}^{-1}\mathrm{Diag}(\{\hat{\Omega}_i^2\}_{i=1}^n)\hat{\mathscr{H}}^{-1}$ and $\hat{\Sigma}_u = \mathrm{Diag}(\{b_i/(\beta_i^\gamma)^2\})\hat{\mathscr{H}}^{-1}\mathrm{Diag}(\{\hat{\Omega}_i^2\}_{i=1}^n)\hat{\mathscr{H}}^{-1}\mathrm{Diag}(\{b_i/(\beta_i^\gamma)^2\})$ be the estimated covariance matrices. We can then construct confidence regions $\mathrm{CR}_\beta$ and $\mathrm{CR}_u$ as above.

**Bootstrap.** Though not studied in this paper, bootstrap inference is a valuable alternative. In a bootstrap procedure, one samples the items with replacement and solve the EG programs (S-EG) or (S-DEG). By repeating this we obtain a sequence of estimates say $\beta^1, \ldots, \beta^B$, where $B$ is the number of bootstrap batches. Then a confidence interval is constructed by the sample quantiles of the sequence $\{\beta^k\}_{k \in [B]}$.

## 6 EXPERIMENTS

We conduct experiments to validate the theoretical findings, namely, the convergence of $\mathrm{NSW}^\gamma$ to $\mathrm{NSW}^*$ (Theorem 9) and CLT (Eq. (2)). All figures can be found in Appendix O.

**Verify convergence of NSW to its infinite-dimensional counterpart in a linear Fisher market.** First, we generate an infinite-dimensional market $\mathscr{M}_1$ of $n = 50$ buyers each having a linear valuation $v_i(\theta) = a_i\theta + c_i$ on $\Theta = [0,1]$, with randomly generated $a_i$ and $c_i$ such that $v_i(\theta) \geq 0$ on $[0,1]$. Their budgets $b_i$ are also randomly generated. We solve for $\mathrm{NSW}^*$ using the tractable convex conic formulation described in Gao and Kroer (2022, Section 4). Then, following Section 2.2, for the $j$-th ($j \in [k]$) sampled market of size $t$, we randomly sample $\{\theta_j^{t,\tau}\}_{\tau \in [t]}$ uniformly and independently from $[0,1]$ and obtain markets with $n$ buyers and $t$ items, with individual valuations $v_i(\theta_j^{t,\tau}) = a_i\theta_j^{t,\tau} + c_i$, $j \in [t]$. We take $t = 100, 200, \ldots, 5000$ and $k = 10$. We compute their equilibrium Nash social welfare, i.e., $\mathrm{NSW}^\gamma$, and their means and standard errors over $k$ repeats across all $t$. As can be seen from Fig. 2, $\mathrm{NSW}^\gamma$ values quickly approach $\mathrm{NSW}^*$, which align with the a.s. convergence of $\mathrm{NSW}^\gamma$ in Theorem 9. Moreover, $\mathrm{NSW}^\gamma$ values increase as $t$ increase, which align with the monotonicity observation in the beginning of Section 4.2.

**Verify asymptotic normality of NSW in a linear Fisher market.** Next, for the same infinite-dimensional market $\mathscr{M}_1$, we set $t = 5000$, sample $k = 50$ markets of $t$ items analogously, and compute their respective $\mathrm{NSW}^\gamma$ values. We plot the enpirical distribution of $\sqrt{t}(\mathrm{NSW}^\gamma - \mathrm{NSW}^*)$ and the probability density of $\mathcal{N}(0, \sigma_{\mathrm{NSW}}^2)$, where $\sigma_{\mathrm{NSW}}^2$ is defined in Theorem 4.[4] Theorem 4 shows that $\sqrt{t}(\mathrm{NSW}^\gamma - \mathrm{NSW}^*) \xrightarrow{\mathrm{d}} \mathcal{N}(0, \sigma_{\mathrm{NSW}}^2)$. As can be seen in Fig. 5, the empirical distribution is close to the limiting normal distribution. A simple Kolmogorov-Smirnov test shows that the empirical distribution appears normal, that is, the alternative hypothesis of it not being a normal distribution is not statistically significant. This is further corroborated by the Q-Q plot in Fig. 4, as the plots of the quantiles of $\sqrt{t}(\mathrm{NSW}^\gamma - \mathrm{NSW}^*)$ values against theoretical quantiles of $\mathcal{N}(0, \sigma_{\mathrm{NSW}}^2)$ appear to be a straight line.

**Verify NSW convergence in a multidimensional linear Fisher market.** Finally, we consider an infinite-dimensional market $\mathscr{M}_2$ with multidimensional linear valuations $v_i(\theta) = a_i^\top\theta + c_i$, $a_i \in \mathbb{R}^{10}$. We similarly sample markets of sizes $t = 100, 200, \ldots, 5000$ from $\mathscr{M}_2$, where the items $\theta_j^t$, $j \in [k]$ are sampled uniformly and independently from $[0,1]^{10}$. As can be seen from Fig. 2, $\mathrm{NSW}^\gamma$ values increase and converge to a fixed value around $-1.995$. In this case, the underlying true value $\mathrm{NSW}^*$ (which should be around $-1.995$) cannot be captured by a tractable optimization formulation.

---

[4]To compute $\sigma_{\mathrm{NSW}}^2$, we use the fact that $p^* = \max_i \beta_i^* v_i(\theta)$ is a piecewise linear function, since $v_i$ are linear. Following (Gao and Kroer, 2022, Section 4), we can find the breakpoints of the pure equilibrium allocation $0 = a_0 < a_1 < \cdots < a_{50} = 1$, and the corresponding interval of each buyer $i$. Then, $\int_0^1 (p^*(\theta))^2 dS(\theta)$ amounts to integrals of quadratic functions on intervals.

ACKNOWLEDGMENTS

This research was supported by the Office of Naval Research Young Investigator Program under grant N00014-22-1-2530, and the National Science Foundation award IIS-2147361.

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
