# OpenReview forum: "Statistical Inference for Fisher Market Equilibrium"
_ICLR.cc/2023/Conference — ICLR 2023 poster_

### Official Review · Reviewer_CZ5S · 2022-10-23

**Confidence:** 2
**Correctness:** 4
**Technical Novelty And Significance:** 4
**Empirical Novelty And Significance:** Not applicable
**Recommendation:** 8

**Clarity, Quality, Novelty And Reproducibility:**

Clarity: Good.
Quality: The writing can be improved to improve the readability for general machine learning researchers.
Novelty: Strong.
Reproducibility: Good. The code for the numerical experiments was provided.


**Strength And Weaknesses:**

Strength:

1. Despite numerous algorithmic results available for computing Fisher market equilibria, limited statistical inference result was available for quantifying the randomness of market equilibrium.

2. These statistical inference results (CLT and confidence intervals) are highly non-trivial. One technical challenge is that the dual objective is potentially not twice differentiable and hence it requires new technical tools. So the technical novelty is strong.


Weaknesses:


1. Writing: The writing of the paper can be further improved.

(1) The current version assumes that readers have good knowledges of Fisher markets and market equilibrium. However, this may not be the case for general ICLR readers. It would be helpful to add some background on these topics and provide some real examples of Fisher markets. In addition, some parts of Gao and Kroer (2022) can be included in the appendix of this paper.

(2) The whole paper is very technical and is hard to follow. It would be helpful to check if some of the math notation and math definition can be simplified or moved to the Appendix. The theoretical results include consistency, finite-sample bounds, CLT, and finally variance estimation for constructing confidence intervals. In my personal opinion, the most interesting part is the CLT and confidence interval parts, followed by the finite-sample bounds. The consistency result is of less interests and can be inferred from the finite-sample bounds. It might be helpful to list key theoretical results in the main paper to save space for additional explanations. In addition, the whole Section 5 on the extension to quasilinear fisher market can be moved to the appendix. This section could also be added in a journal version. On the other hand, some of the numerical experiments in Section M of the appendix may be moved to the main paper.


2. Motivation. It would be helpful to add more justifications on why the statistical inference for linear Fisher market equilibrium is useful. There are two concerns.

(1) The authors claimed in the Introduction that statistical inference can be used to "quantify the variability of the resources/fairness/efficiency metrics of interest in the long run". But why do we want to quantify the variability of these metrics in the long run? The example of Internet ad auction markets used in the Introduction did not fully answer this question. Why should an internet company care about the statistical inference of these metrics considered in this paper?

(2) This paper considers a linear Fisher markets where the valuation for each buyer is linear. Although this helps the theoretical developments, such assumptions might be restrictive in practice. Interestingly, the ad auction markets example (Conitzer et al., 2022a) provided in the Introduction of this paper actually corresponds to a quasilinear Fisher market. More justifications on the linear fisher markets would be needed.


3. Real data analysis. This is related to the previous comment. It would be more convincing if a real data study is added.


4. Constructing confidence intervals for the pacing multipliers and the utilities. In Section 4.3 Inference, Theorem 8 only contains the inference result for Nash social welfare because the estimation of the variance for $\beta$ and $u$ is more complicated. In the later case, the plug-in estimator is no longer suitable. Can we use Bootstrap method to estimate the variance?


**Summary Of The Paper:**

This paper considers statistical inference in linear Fisher markets. In their framework, a market formed by a finite number of items sampled from an underlying distribution is observed and their goal is to infer several important equilibrium quantities (individual utilities, pacing multipliers, and social welfare) of the underlying long-run market.



**Summary Of The Review:**

This paper considers statistical inference in linear Fisher markets. Finite-sample bounds, CLT, and statistical inference of a few metrics are derived. The technical contribution is strong as the inference for Fisher markets is non-trivial. However, there are a few major concerns listed in the weakness part. If the authors can nicely address these weakness parts, I am willing to increase my rating later.

---

> ### Author Response · Authors · 2022-11-09
> **Paper reorg and model meaning in tech applications**
>
> > Writing: The writing of the paper can be further improved.
> (1) The current version assumes that readers have good knowledge of Fisher markets and market equilibrium. However, this may not be the case for general ICLR readers. It would be helpful to add some background on these topics and provide some real examples of Fisher markets. In addition, some parts of Gao and Kroer (2022) can be included in the appendix of this paper.
> (2) The whole paper is very technical and is hard to follow. It would be helpful to check if some of the math notation and math definition can be simplified or moved to the Appendix. The theoretical results include consistency, finite-sample bounds, CLT, and finally variance estimation for constructing confidence intervals. In my personal opinion, the most interesting part is the CLT and confidence interval parts, followed by the finite-sample bounds. The consistency result is of less interest and can be inferred from the finite-sample bounds. It might be helpful to list key theoretical results in the main paper to save space for additional explanations. In addition, the whole Section 5 on the extension to quasilinear fisher market can be moved to the appendix. This section could also be added in a journal version. On the other hand, some of the numerical experiments in Section M of the appendix may be moved to the main paper.
>
> We sincerely thank you for the detailed editorial comments. Per your advice, we will edit the paper as follows (we will post the edited version before Nov 18)
>
> - Add more introductory texts for Fisher markets
> - Add a Guide to Practitioners section describing how to construct our estimators, variances, etc. at a high level, and then describe the more mathematical results afterward (with some of it potentially going to the appendix)
> - Move consistency results to the appendix.
> - Move numerical studies to the main text.
> - For the extension to quasilinear markets, we think those results are quite useful since they cover the ad auction setting, but we agree that for space reasons they can be moved to the appendix.
>
> > The authors claimed in the Introduction that statistical inference can be used to "quantify the variability of the resources/fairness/efficiency metrics of interest in the long run". But why do we want to quantify the variability of these metrics in the long run? The example of Internet ad auction markets used in the Introduction did not fully answer this question. Why should an internet company care about the statistical inference of these metrics considered in this paper?
>
>
> For ad platforms, individual utility reflects, for example, the value generated to advertisers. A confidence interval on this quantity can be provided by the ad platform to the advertisers to better inform advertisers’ decision making. Moreover, this can be helpful when evaluating changes to the platform. Pacing multiplier reflects the advertisers’ bidding strategy, a confidence interval on this quantity could help the ad platform predict its clients bidding behavior. Similarly, ad platforms will obviously be interested in estimating revenue variability, and testing how it is impacted by changes to the platform. Finally, Nash social welfare measures the efficiency of the whole ad platform. When compared on a year-by-year basis, it directly reflects how well the ad platform serves its clients.
>
> For the *linear* Fisher market, we can use this to model other internet applications. Resource or task allocation systems are common in internet companies. For example, in a job recommendation platform, we can model it as a market where we distribute job posts to viewers (job seekers). In this context, social welfare measures generally the efficiency of the job recommendation system. Individual utilities track how satisfied the job post creators are with the extent to which their job posts are being recommended. Although not studied in this paper, the allocation vector reflects the rate of recommendation of various types of jobs. When focusing on the job recommendation rates of a subgroup of buyers (say under-represented groups), being able to do inference about the allocation vector would help achieve fairness guarantees.
>
> As another example, in the robust content review problem, we are faced with the task of filtering several types of harmful social media content (e.g., fake news, impersonation, hate speech, . . . ). The goal is to allocate review time to the content categories in a way that satisfies all forecasted review amounts and then allocate the excess reviewing capacity across the content types to be robust to variations from the forecast. In this setting, the content categories are “buying” review time. In that case, we might be interested in estimating the utility of each content category, which translates into statements about the quality of the content reviewed.

---

> > ### Comment · Reviewer_CZ5S · 2022-11-17
> > **Increased score.**
> >
> > Thanks for the clarification. I have increased the score.

---

> ### Author Response · Authors · 2022-11-09
> **Justify linear Fisher market and other comments**
>
> > This paper considers linear Fisher markets where the valuation for each buyer is linear. Although this helps the theoretical developments, such assumptions might be restrictive in practice. Interestingly, the ad auction markets example (Conitzer et al., 2022a) provided in the Introduction of this paper actually corresponds to a quasilinear Fisher market. More justifications on the linear fisher markets would be needed.
> Understanding the linear Fisher market is fundamental for the quasilinear market due to their similarity in the market equilibrium structure (first-order conditions). For consistency results and high-probability bounds, the derivation of these results for the quasilinear market will be the same as the linear market. Moreover, our paper explores revenue inference with a finite-sample high-probability bound under the quasilinear market. It is also worth noting that the linear model is the standard model in the academic literature, whereas the QL model is a much later extension, so we believe that from an academic perspective, it is more natural to present the linear case results; especially since those are slightly more readable while conveying many of the same ideas.
>
> Moreover, note that in the examples we gave to your point (1) above, two of them are for the linear case: when using Fisher markets as a way to fairly allocate content recommendations (e.g. job ads), or in the content review problem, the appropriate model is the linear model, not the quasilinear model.
>
> > Real data analysis. This is related to the previous comment. It would be more convincing if a real data study is added.
>
> (This response was also sent to reviewer CZ5S as they brought up the same concern)
>
> We agree that a real-world dataset with experiments would be nice to have. However, we would like to point out that a theoretical framework for reasoning about statistical inference is required before doing real-world experiments. As far as we know, no paper has ever studied statistical inference in Fisher markets before. We believe that developing the theoretical framework is an important first step in this process in its own right, and we do not think that real-world experiments should be required when we are initiating the study of this problem. In fact, we believe that there is quite a lot of interesting work to be done on the theoretical and applied front before real-world experiments start to become a first-order concern for this problem.
>
> > Constructing confidence intervals for the pacing multipliers and the utilities. In Section 4.3 Inference, Theorem 8 only contains the inference result for Nash social welfare because the estimation of the variance for β and u is more complicated. In the later case, the plug-in estimator is no longer suitable. Can we use the Bootstrap method to estimate the variance?
>
> To clarify, in the paper we do discuss ways to do inference on beta and u under extra conditions, which are in the appendix. Under a stronger form of twice differentiability conditions, the asymptotic variance expressions for beta and u are simplified, facilitating inference. In another scenario with merely the twice differentiability assumption (See appendix J for many realistic examples of this assumption being true), where we have access to the values of agents on the realized items (note we are not requiring access to the whole value function $v_i(\cdot)$), we can use a numerical difference method to estimate the Hessian matrix and thus the asymptotic variance. In the revised version, we are adding a Guide to Practitioners section to describe how to do this.
>
> We agree that bootstrap is a valuable alternative to the numerical difference method discussed in the appendix. Both of these methods (numerical difference and bootstrap) require additional knowledge beyond simply the prices/utilities/betas; they also require the valuations on the realized goods. This is fairly realistic though. We will add a discussion of this in the Guide to Practitioners too.

---

### Official Review · Reviewer_PJb7 · 2022-10-25

**Confidence:** 2
**Correctness:** 3
**Technical Novelty And Significance:** 2
**Empirical Novelty And Significance:** 2
**Recommendation:** 6

**Clarity, Quality, Novelty And Reproducibility:**

Needs experimental design and publicly available data set to evaluate the proposed methodology or at the least a section on how to leverage the methodology in a real world application.
The authors had provided detailed materials to reproduce the results.


**Strength And Weaknesses:**

Strengths
The article is well written with sufficient support from literature; the proposed method and its limitations are well written and gives good readability.
Results and conclusion adhere to the style of writing and provides better readability and clarity.
Detailed theoretical and empirical evaluation


**Summary Of The Paper:**

The authors have proposed a statistical framework based on the concept of infinite dimensional Fischer markets. This study attempted a marked formed by the finite number of items sampled from an distribution to infer several equilibrium quantities such as individual utilities, social welfare, and pacing multipliers of the underlying long-run market.

**Summary Of The Review:**

This paper provides good readability but it has gone in depth to analyze/explain the theoretical algorithm in detail on the fisher market equilibrium.
This article is an incremental work of Gao and Kroer (2022) and extended the Fischer market model and proposed a statistical model based on their infinite-dimensional Fischer Market.

---

> ### Author Response · Authors · 2022-11-09
> **Thank you for your suggestion. We will add a Guide to Practitioners section.**
>
> > Needs experimental design and publicly available data set to evaluate the proposed methodology or at the least a section on how to leverage the methodology in a real world application.
>
> We agree with the reviewer that a real-world dataset with experiments would be nice to have. However, we would like to point out that a theoretical framework for reasoning about statistical inference is required before doing real-world experiments. As far as we know, no paper has ever studied statistical inference in Fisher markets before. We believe that developing the theoretical framework is an important first step in this process in its own right, and we do not think that real-world experiments should be required when we are initiating the study of this problem. In fact, we believe that there is quite a lot of interesting work to be done on the theoretical and applied front before real-world experiments start to become a first-order concern for this problem.
>
> That said, you are right that it would be useful to have a methodology section describing in more practical terms how to apply our framework. We will add a Guide to Practitioners section in the revised version. Second, we point out that in the appendix we provide preliminary experiments already. Another reviewer also asked us to feature those experiments in the body of the paper, and so we will revamp the presentation to add those there. Please see our reply to reviewer CZ5S for the specific plan on how we will do this.
>
> > This article is an incremental work of Gao and Kroer (2022) and extended the Fischer market model and proposed a statistical model based on their infinite-dimensional Fischer Market.
>
> We strongly disagree that our work is an incremental work on top of Gao & Kroer. We agree that our paper *relies* on that paper, since we propose a statistical framework which is based on estimating an underlying market that comes from their infinite-dimensional Fisher market model. However, that previous paper has no discussion of statistical inference whatsoever, which is the primary topic of this paper. For that reason, it is hard for us to see how the reviewer could possibly think that our work is incremental based on that prior work. Could you please point to specific examples of results in our paper and describe how those results are easy to show based on the results in Gao & Kroer?
>
> Moreover, we want to reemphasize that our paper is the first to ever study statistical inference in Fisher markets. Thus, it seems hard to argue that we could be incremental on top of any prior paper, since our topic has not even been studied before.

---

### Official Review · Reviewer_8p3H · 2022-10-26

**Confidence:** 3
**Correctness:** 4
**Technical Novelty And Significance:** 3
**Empirical Novelty And Significance:** Not applicable
**Recommendation:** 8

**Clarity, Quality, Novelty And Reproducibility:**

The paper builds on an emerging line of work that brings statistical considerations into classical markets. The study of how well observed market quantities approximate the true market quantities is natural and interesting. The main weakness is it is not clear if the setup for the observed market is well-motivated.

The paper is very well-written and the statistical results are clearly explained. The background section (Section 2) is useful for a machine learning audience.

**Strength And Weaknesses:**

Strengths
- The paper bridges linear Fisher markets with statistical inference in an interesting manner. The results on fundamental statistical properties—consistency and asymptotic normality—in the context of a classical market are insightful.
- The technical results in the paper—especially the consistency results and finite sample bounds as well as asymptotic normality—are mathematically clean and clearly explained.

Weaknesses
- The main weakness of this paper is in the motivation for the setup. In particular, the paper assumes the observed data comes from a market equilibrium from a finite item sample of the underlying continuum market. However, the paper does not provide much motivation for this modeling choice. Why it is not more natural for the learner to have data from a noisy version of the continuum item market, e.g. the equilibria formed under random perturbations in the values of agents?
- What are the statistical properties of other measures of social welfare besides Nash social welfare?

**Summary Of The Paper:**


The paper proposes a statistical framework to estimate linear Fisher markets with a continuum distribution over items from observed markets with finitely many items drawn from the distribution. A linear Fisher market consists of a continuum set of items with predetermined supplies and a finite set of buyers (each with a potentially different budget and different valuations for items). The market outcome is an allocation of items to buyers along with prices. The goal is to estimate certain quantities at equilibrium in the continuum market: (1) individual utilities received by buyers, (2) pacing multipliers (defined by the budget divided by the utility), and (3) Nash social welfare. The paper examines the statistical properties of equilibrium market quantities from the observed market with finitely many items as estimators for the corresponding quantities of the continuum market. The main results are showing the following: (A) consistency of these estimators and finite sample bounds for the Nash social welfare, the individual utilities, and the pacing multipliers, (B) convergence of approximate market equilibria, and (C)  asymptotic normality of the Nash social welfare.

**Summary Of The Review:**

The paper investigates statistical considerations that arise in linear Fisher markets and contributes to an emerging line of work bridging statistics and market equilibria. The main weakness is the motivation for how the observed quantities are set up in this paper. Despite this weakness, the paper offers a number of informative insights about the convergence guarantees of linear Fisher markets.

---

> ### Author Response · Authors · 2022-11-09
> **Model motivation**
>
> > Why is it not more natural for the learner to have data from a noisy version of the continuum item market, e.g. the equilibria formed under random perturbations in the values of agents?
>
> Our model can be considered as an even higher-level model for the following two finer-grained models. First, imagine a setting where items are represented by a set of features, and agents have valuations that are functions of those features. In that case, the natural model is that we sample a finite set of items (i.e. feature vectors), e.g., in ad auctions or in recommender system settings. In that case the continuum model reflects the ex-ante expected utilities that we would like to give to agents, whereas the observed market corresponds to a particular realized market . Second, our setup allows us to consider a setting with a finite set of "base goods" where the realized good is a base good with some perturbation applied to it (in other words: we observe a noisy version of the base goods). If there's a continuum of possible perturbations then this corresponds to a continuum model. In this sense, our model is natural. We think this second model is actually quite similar to the model suggested by the reviewer.
>
> We want to remark on the alternative observation model you proposed (“equilibria formed under random perturbations in the values of agents”). First, we don’t think that it is a very realistic model to observe noisy *continuous* versions of the continuum item market. The objects (price function, value function and the allocation) in the continuum item market are infinite dimensional objects (as functions on the whole item set, which can be continuous). We don’t think it is natural, in most settings, to assume that we observe those entire functions (it may be reasonable in something like a repeated cake-cutting setting). However, we do agree that incorporating noisy values in the finite-item case is an interesting setting. As stated above, the continuum model captures the ex-ante fair expected utilities in this case. Second, under certain regularity conditions (see Appendix C) the inferential procedure for social welfare, beta, and u do not even require knowledge of the values on realized items (the $v_i(\theta^\tau)$ variables). In this sense, we are treating values as private information of the agents which the data analyst does not have access to. It is natural to assume that when forming the equilibrium the agents know exactly how much they value the items.
>
> > What are the statistical properties of other measures of social welfare besides Nash social welfare?
>
> We focus on Nash social welfare for the following reasons. First, NSW is scale-free due to taking logs, other social welfare metrics such as $\sum_i u_i ^\alpha $ for some exponent $\alpha > 0$ are not well-defined because it depends on the scaling. To give an extreme example, suppose one of the buyers scales their value by 100 while other buyers have values that sum to 1: the Fisher market equilibrium remains the same, yet if we cared about standard social welfare then we now have to give all the goods to the buyer that scaled their valuation. This is why standard social welfare is typically not studied in these budgeted settings. Another example would be in the ad auction setting: suppose some advertiser has values summing to 1000 dollars, but their budget is 1. In that case, the platform would want to think of the ``effective value’’ of that buyer, which should be on the order of 1 as opposed to 1000. This is why Fisher markets focus on NSW and Pareto optimality.
>
> Secondly, one can extract statements about Pareto optimality based on our results. Since we have convergence for NSW, we will also be "near" some Pareto-optimal utility vector. Finally, we note that the asymptotic variance of NSW has a clean expression, facilitating confidence interval construction.
>
> Thirdly, since we have established the joint asymptotic normality of the utility vector, the asymptotic distribution of any other social welfare that is a smooth function of utility can be derived by the delta method. For example, this could be used to make statements about standard social welfare.
>
> We also discuss the meaning of various metrics (Social welfare, pacing multiplier and utilities) in the context of technology industries, especially Internet advertising, in the response to Reviewer CZ5S.
>
> > correctness: 1
>
> We noticed that the reviewer gave us a correctness score of 1. Was that intentional? Our theories are supported by rigorous mathematical proofs, and there seems to have been no flaws identified by any of the reviewers, so we were surprised to see that since our results all seem to be correct. We kindly ask the reviewer to elaborate on what is meant if you really did mean to give a 1.

---

> > ### Comment · Reviewer_8p3H · 2022-11-16
> > **Updated score**
> >
> > Thanks for the detailed response! I'm satisfied with the authors' responses (especially the justification of considering finite realizations of the market) and have updated my score to reflect this.

---

### Author Response · Authors · 2022-11-09
**To all reviewers**

We have posted responses to each reviewer under their corresponding review. We are now working on incorporating the promised changes in the paper. We hope to post an updated version by the end of the week, but we wanted to post our response ASAP in case the reviewers have additional comments right away.

---

> ### Author Response · Authors · 2022-11-14
> **Rebuttal revision posted**
>
> Dear reviewers,
>
> We revised the paper accordingly. We sincerely thank you for the valuable revision suggestions. Please let us know if there are any remaining issues that make you hesitant to update the ratings.
>
> Concretely, we made the following changes:
>
> - Add more introductory texts for Fisher markets.
> - Add a "mapping model to concrete applications" section.
> - Add a Guide to Practitioners section summarizing how to do inferences about social welfare, utilities, and pacing multipliers in the presence of equilibrium effects.
> - Move numerical studies to the main text.
> - Move consistency results to the appendix.

---

### Decision · Program_Chairs · 2023-01-20

**Decision:**

Accept: poster

**Justification For Why Not Higher Score:**

The contributions are interesting, but maybe a bit narrow compared to other papers.

**Justification For Why Not Lower Score:**

It is a good paper

**Metareview: Summary, Strengths And Weaknesses:**

This is a good paper on fair resource allocation, which an interesting topic with many potential applications.

The contributions might not be mind-blowing, but they are still of quality, putting this paper above the acceptance bar.

**Note From Pc:**

if the above contains the word "oral" or "spotlight" please see: "oral" presentation means -> notable-top-5% and "spotlight" means -> notable-top-25%. As stated in our emails, we are disassociating presentation type from AC recommendations